# Outer membrane vesicles hijack TIM-1 for cellular uptake

Craig R. MacNair[1], Varnesh Tiku[1,2], Shengya Cao[3,4], Ariana D. Sanchez[3], Barath Udayasuryan[1], Katharina Theresa Kroll[1], Adarsh Singh[1,5], Man-Wah Tan[1]*

1 Infectious Diseases and Host-Microbe Interactions Department, Genentech Inc., San Francisco, California, United States of America, 2 Department of Oncology, Gilead Sciences Inc., Foster City, California, United States of America, 3 Microchemistry, Proteomics and Lipidomics, Genentech, San Francisco, California, United States of America, 4 S.C. Translation, Inceptive, Palo Alto, California, United States of America, 5 Meinig School of Biomedical Engineering, Cornell University, Ithaca, New York, United States of America

* tan.man-wah@gene.com

## Abstract

Outer membrane vesicles (OMVs) are nanoscale proteoliposomes shed by Gram-negative bacteria that mediate host-pathogen interactions and hold promise as platforms for vaccines and targeted drug delivery. Despite their biological and translational significance, the cellular mechanisms governing OMV entry into host cells remain poorly understood. Here, we demonstrate that *E. coli* OMVs are internalized by epithelial cells via clathrin-mediated, receptor-dependent endocytosis. Using a high-throughput screen of over 1,500 human single-pass transmembrane proteins, we identify T-cell immunoglobulin and mucin-domain 1 (TIM-1) as a strong OMV-binding receptor. Functional validation revealed that TIM-1 overexpression markedly increased OMV uptake, whereas TIM-1 knockout and antibody-mediated blockade significantly impaired internalization across multiple cell lines. Mechanistic studies demonstrate that TIM-1 binds to lipopolysaccharide (LPS) on the OMV surface via its phosphatidylserine-binding domain. Uptake of OMVs by TIM-1 triggers proinflammatory cytokine production which can be reduced by preventing this interaction. Additionally, OMVs from multiple bacterial species hijack TIM-1 for entry, making it an intriguing antivirulence strategy. Our findings establish TIM-1 as a critical host receptor mediating OMV uptake and provide a novel approach to modulate vesicle-driven pathogenesis and enhance OMV-based therapies.

## Author summary

Gram-negative bacteria constitutively shed outer membrane vesicles (OMVs), which are nanoscale lipid particles that transport bacterial proteins and genetic material to host cells. These vesicles play an important role in bacterial pathogenesis and host-immune modulation. However, the mechanisms by which

**Data availability statement:** All relevant data are within the manuscript and its Supporting Information files.

**Funding:** This study was supported by Genentech, Inc. Genentech was involved, in conjunction with the authors, in the study design, data interpretation and the decision to publish the manuscript. All authors were funded by Genentech at the time of this work.

**Competing interests:** I have read the journal's policy and the authors of this manuscript have the following competing interests: All authors were employees of Genentech Inc. at the time of this work.

human cells internalize them have remained largely uncharacterized. In this study, we identified T-cell immunoglobulin and mucin-domain 1 (TIM-1) as a major receptor driving OMV internalization. Our findings demonstrate that TIM-1 recognizes specific lipids on the OMV surface, triggering a cellular uptake process known as clathrin-mediated endocytosis. By using genetic engineering and neutralizing antibodies, we confirmed that TIM-1 is a critical determinant of OMV entry and the subsequent host response. These results elucidate a fundamental and conserved mechanism of host-pathogen interaction. Furthermore, because OMVs are currently being developed as platforms for new vaccines and targeted drug delivery, understanding this entry pathway provides a molecular framework for refining these biotechnological tools to improve human health.

## Introduction

Outer membrane vesicles (OMVs) are constitutively produced during growth and upregulated in response to environmental stressors [1,2]. Ranging in size from 20 to 400 nm in diameter, OMVs carry a sampling of periplasmic contents enclosed by an outer membrane-derived bilayer containing lipopolysaccharide (LPS) on the outer leaflet, phospholipids on the inner leaflet and membrane-associated proteins [3]. This structure makes OMVs a stable delivery system for a broad range of virulence factors, including toxins, adhesins, and pathogen-associated molecular patterns (PAMPs) [4,5].

In the absence of live bacteria OMVs are sufficient to cause disease and even lethality. For example, *Escherichia coli* OMVs can induce mortality in a mouse model of sepsis [6], and Shiga toxin containing OMVs from Enterohemorrhagic *E. coli* (EHEC) cause endothelial injury and hemolytic uremic syndrome [7]. While some effects are driven by interactions restricted to host cell surfaces [8–11], most OMV-mediated pathogenesis requires vesicle entry. OMVs deliver peptidoglycan into epithelial cells, activating NOD1 and inducing IL-8, IL-6, and NF-κB secretion [12]. Delivery of intracellular LPS by OMVs upregulates IL-1β and triggers pyroptosis [13]. Many virulence proteins, such as cytolethal distending toxin [14], OmpA [15], PorB [16], and vacuolating cytotoxin A (VacA) [17], can leverage OMV packaging to improve entry into host cells and reach their intracellular targets.

In addition to playing key roles in host-pathogen interactions, OMVs serve as an attractive biotechnology delivery platform. Their intrinsic immunogenicity and capacity for multivalent antigen display have enabled the development of effective vaccines, including clinically approved formulations for serogroup B *Neisseria meningitidis* [18–20]. OMVs are also being developed as targeted drug delivery vehicles. They can be engineered to encapsulate small molecules, enzymes, or nucleic acids and to display surface ligands that enrich delivery to selected tissues [21,22]. This approach has shown particular promise in immuno-oncology indications, [23,24] where OMVs designed to accumulate in HER2 + tumors and deliver a therapeutic siRNA reduced tumor growth in a murine xenograft model [25].

Despite the importance of OMV internalization into host cells for both biotechnology applications and OMV-mediated pathogenesis, the mechanisms behind cellular entry remain poorly characterized. Most studies have relied on broad endocytosis inhibitors that implicate clathrin-mediated uptake, macropinocytosis, or lipid-raft-dependent entry, as dominant routes of uptake with little mechanistic resolution [5,26,27]. OMVs can also enter host cells through direct membrane fusion, a process where the vesicle's lipid bilayer merges with the host plasma membrane to deliver internal cargo [5,26,27]. OMV features such as size [28], LPS structure [29], and surface proteins [29] correlate with changes in uptake, but how these features control cellular entry remains unclear. A few ligand-based interactions involving unique bacterial surface proteins have been proposed [29]. For example, the virulence factor VacA found on *Helicobacter pylori* OMVs engages its receptor LRP1, which promotes internalization [30]. Furthermore, in *Vibrio cholerae*, cholera toxin-containing OMVs utilize ganglioside GM1 to facilitate entry, while Shiga toxin-bearing OMVs can engage globotriaosylceramide (Gb3) to initiate internalization [29]. However, isogenic bacterial strains lacking these proteins still produce OMVs that enter host cells, underscoring our incomplete understanding of internalization.

Here, we set out to identify the molecular determinants of OMV uptake. Screening more than 1,500 human single-pass transmembrane proteins revealed several receptors that bind OMVs. Complementary gain- and loss-of-function studies showed that the T-cell immunoglobulin and mucin domain 1 (TIM-1) mediates OMV entry across multiple cell types. We found that OMVs from diverse Gram-negative bacteria, including *E. coli*, *Pseudomonas aeruginosa*, *Salmonella* Typhimurium, and *Fusobacterium nucleatum*, hijack TIM-1 to promote internalization. Mutational and binding analyses demonstrated that this interaction occurs between LPS on the OMVs and the phosphatidylserine-binding pocket of TIM-1. Entry via TIM-1 increased proinflammatory cytokine release and blocking TIM-1 reduced OMV-driven pathology. Together, these findings establish TIM-1 as a key receptor for OMV uptake and lay the groundwork for understanding mechanisms of OMV-mediated microbe-host interactions, as well as receptor-guided strategies to modulate OMV delivery.

## Results

### Receptor-mediated endocytosis drives OMV uptake

To establish the molecular determinants of mammalian cells that mediate OMV uptake, we first looked to identify OMVs that are predominantly internalized through a receptor-mediated route of endocytosis. Prior studies suggest that OMVs from the K-12 strain of *E. coli* enter non-phagocytic cells through clathrin-dependent, receptor-mediated endocytosis [29]. To monitor OMV uptake, we exposed A549 lung epithelial cells to *E. coli* K-12 vesicles stained with the fluorescent lipophilic dye DiO and measured uptake using flow cytometry. Trypan blue was used to quench surface fluorescence, enriching for the internal OMV signal [30]. Using this assay, we observed a time-dependent increase in fluorescence (Fig 1A) consistent with the reported active uptake of these OMVs [29].

We next tested the impact of global and pathway-specific perturbations of endocytosis on OMV internalization (Fig 1B). Inhibition of endocytosis by incubation at 4°C and targeting dynamin-dependent endocytosis with the GTPase inhibitor dynasore nearly abolished uptake, reducing OMV internalization in A549 cells by approximately 96% and 93%, respectively (Fig 1C). To assess individual pathways of endocytosis, we treated A549 cells with inhibitors of macropinocytosis (blebbistatin), lipid-raft mediated (filipin), caveolin-mediated (filipin) and clathrin-dependent endocytosis (chlorpromazine). Of the pathway-specific inhibitors chlorpromazine had the largest effect, decreasing OMV uptake by ~92%. Lipid-raft inhibition resulted in a decrease of ~15%, whereas inhibition of macropinocytosis had no effect on OMV uptake when compared to a vehicle control (Fig 1C). We also assessed the impact of perturbing endocytosis on OMV entry in the intestinal epithelial cell line Caco-2. Here, we observed a similar pattern of inhibition as in A549 cells, with 4°C, dynasore, and chlorpromazine yielding the strongest reductions in OMV internalization (Fig 1D). These findings indicate that the *E. coli* OMVs are taken up by epithelial cells primarily via clathrin-mediated, receptor-dependent endocytosis, suggesting the presence of unidentified receptors responsible for OMV capture and entry.

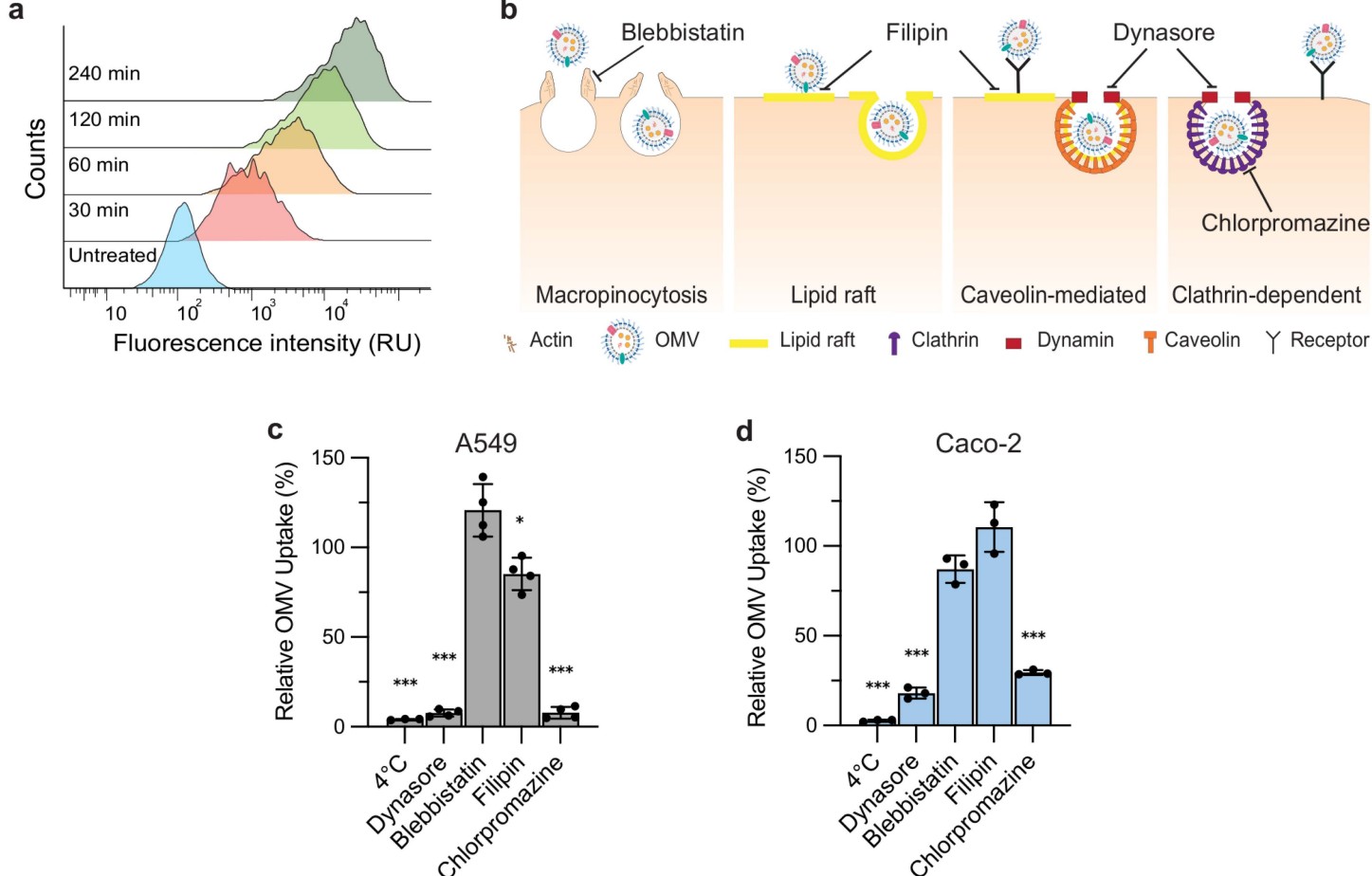

**Fig 1. *E. coli* K-12 OMVs enter epithelial cells mainly via clathrin-mediated endocytosis. (A)** Flow cytometry histograms showing time-dependent increases in fluorescence relative units (RU) of A549 cells incubated with DiO-labeled OMVs for the indicated times. **(B)** Schematic of potential routes of OMV entry and inhibitors of these pathways. Blebbistatin inhibits myosin II–driven contractility, impairing macropinocytosis. Filipin sequesters membrane cholesterol and disrupts lipid rafts and caveolae, reducing raft- or caveolin-dependent OMV uptake. Dynasore blocks dynamin-dependent vesicle scission, inhibiting both clathrin- and caveolin-mediated endocytosis. Chlorpromazine prevents clathrin lattice assembly/disassembly, blocking clathrin-mediated endocytosis. **(C)** Relative OMV uptake in A549 cells after 2 hours under the indicated conditions. Values are normalized to a vehicle control (= 100%) and analyzed using one-sample t-test vs 100% (two-sided, * $P<0.05$, *** $P<0.001$). Bars show mean±s.d. (n=4). **(D)** The same perturbations and normalization in Caco-2 cells (n=3) as in (**C**).

## Human single-pass transmembrane proteins bind OMVs

Using a high-throughput receptor screening platform containing Fc-tagged ectodomains representing over 1,500 human single-pass transmembrane (STM) proteins, we sought to identify OMV binders (Fig 2A). Nanoluciferase-containing OMVs were incubated with individual STM proteins, washed, and luminescence intensity recorded as a proxy for binding. Twenty-six STM proteins were found to reproducibly bind OMVs (Fig 2B and Table A in S1 Text). Gene Ontology (GO) analysis of these hits showed enrichment for phosphatidylserine (PS) binding, modified amino acid binding, heparin binding, glycosaminoglycan binding, and sulfur compound binding, consistent with proteins that bind anionic ligands (Table B in S1 Text). Notably, OMVs are anionic due to the presence of lipopolysaccharide. We prioritized five hits for follow-up analysis based on their strong luminescent signal and/or known roles in receptor-mediated endocytosis: WW domain-binding protein 1 (WBP1), amyloid precursor-like protein 2 (APLP2), T-cell immunoglobulin and mucin domain 1 (TIM-1), T-cell

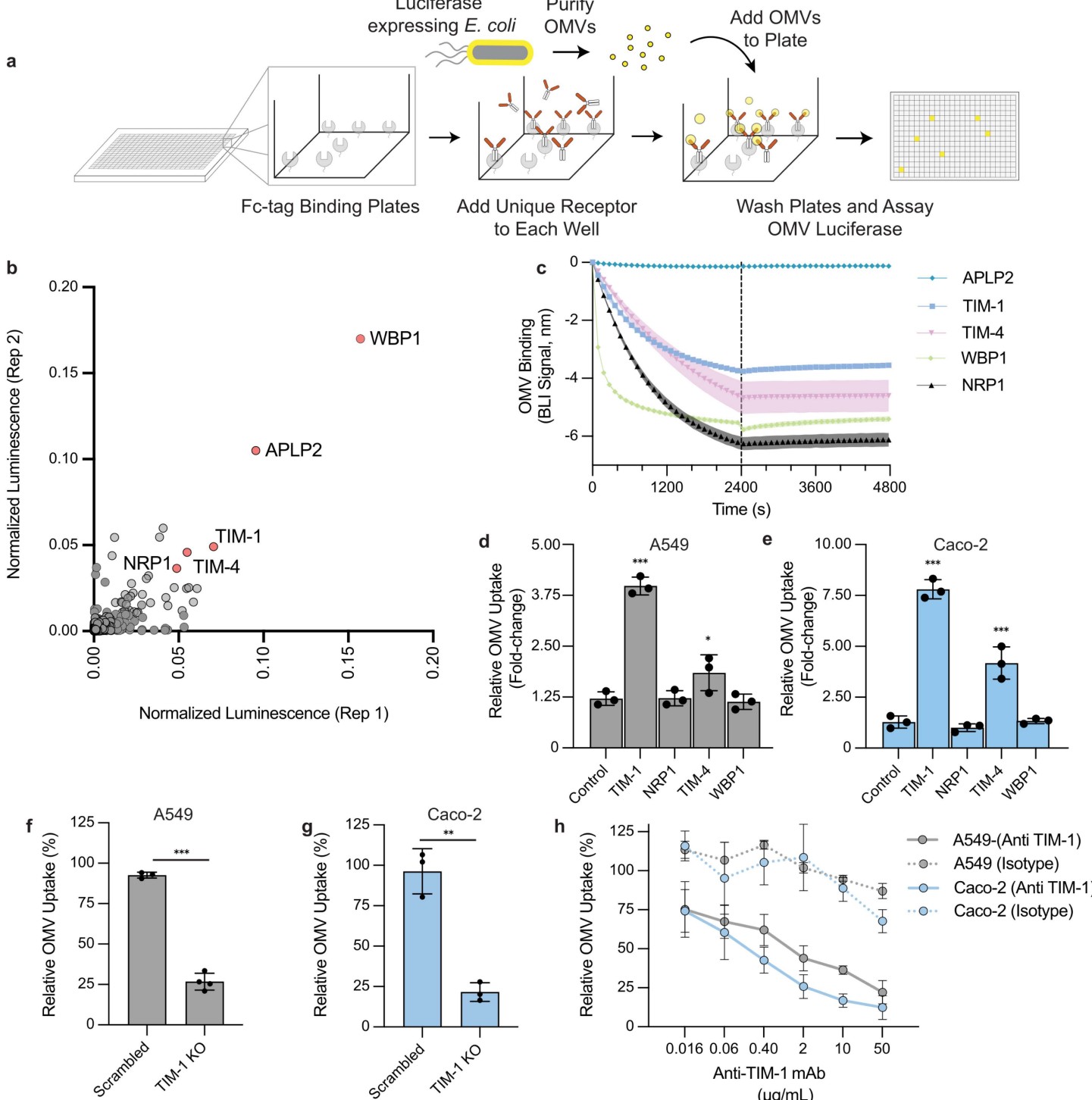

**Fig 2. Single-pass transmembrane receptors bind OMVs, and TIM-1 drives vesicle uptake. (A)** Schematic of the OMV ectodomain binding screen. Purified Fc-tagged human single-pass transmembrane (STM) ectodomains were arrayed, incubated with NanoLuc-containing OMVs, washed, and luminescence assayed as a proxy for binding. **(B)** Replica plot of normalized luminescence for the STM ectodomains. Highlighted (red) STM proteins TIM-1, TIM-4, NRP1, APLP2, WBP1 were selected as OMV binders and advanced for further validation. **(C)** Biolayer interferometry (BLI) of OMVs binding to immobilized STM ectodomains. Traces show a strong negative association signal and slow dissociation, indicating binding for TIM-1, TIM-4, NRP1,

and WBP1 but not APLP2. Each solid line represents the mean and the shaded band is ± s.d. of at least two biological replicates **(D–E)** Overexpression alters OMV internalization as measured by flow cytometry after 2 hours of DiO-labeled OMV exposure. Relative uptake was normalized to wildtype cells (=1). **(D)** A549 (n = 3). **(E)** Caco-2 (n = 3). Bars represent the normalized mean to a wildtype control ± s.d. *** P < 0.001, * P < 0.05, one-way ANOVA with Dunnett's multiple comparisons vs vector. **(F–G)** TIM-1 knockout reduces OMV uptake compared to a scrambled control with relative uptake normalized to wildtype cells (= 100%) **(F)** A549 (n = 3) and **(G)** Caco-2 (n = 3). Bars represent mean ± s.d. *** P < 0.001, ** P < 0.01, two-sided unpaired t-test vs scrambled. **(H)** Anti–TIM-1 monoclonal antibody (mAb) inhibits OMV uptake in A549 and Caco-2 cells in a dose-dependent manner (n = 3) compared to an isotype control (n = 2). Data is normalized to a vehicle control and lines represent the mean ± s.d.

immunoglobulin and mucin domain 4 (TIM-4), and neuropilin-1 (NRP1). For all receptors except APLP2, biolayer interferometry (BLI) confirmed the binding interaction, showing a robust negative signal during the association phase and slow dissociation, indicative of a high-avidity interaction (Fig 2C). A negative signal in the BLI assay for OMVs is expected due to their large size [31, 32].

## TIM-1 modulates uptake of OMVs

Having confirmed that TIM-1, TIM-4, NRP1, and WBP1 bind OMVs, we tested whether they mediate vesicle internalization by overexpressing each protein in A549 (Fig 2D) and Caco-2 (Fig 2E) cells. Overexpression allows us to determine the sufficiency of each protein in OMV uptake without concern for redundant mechanisms of entry. Caco-2 and A549 cell lines are reported to endogenously express NRP1, WBP1, TIM-1 and low levels of TIM-4 [33]. Overexpression of NRP1 and WBP1 did not increase internalization relative to a vector control (Fig 2D and 2E). In contrast, TIM-1 and TIM-4 markedly enhanced uptake in both cell types. TIM-1 overexpression significantly increased uptake ~4.0-fold in A549 cells and ~7.8-fold in Caco-2 cells. TIM-4 overexpression significantly increased uptake ~1.8-fold in A549 cells and ~4.2-fold in Caco-2 cells. TIM-1 and TIM-4 are closely related TIM family receptors that both recognize phosphatidylserine through a conserved binding pocket. Unlike TIM-4, which functions mainly as a tether for other phagocytic machinery [34, 35], TIM-1 can directly facilitate receptor-mediated endocytosis [36, 37]. We therefore focused subsequent follow-up and mechanistic studies on TIM-1 as a receptor that could singularly drive OMV uptake.

To confirm that TIM-1 mediates OMV internalization, we next knocked out and selected for TIM-1-negative A549 and Caco-2 cells. Loss of TIM-1 decreased uptake by ~75% in A549 (Fig 2F) and ~80% in Caco-2 (Fig 2G) cells when compared to a scrambled control. We also evaluated whether antibody blockade of TIM-1, using a monoclonal antibody (mAb) previously shown to impair TIM-1-dependent viral entry [38], is sufficient to reduce OMV internalization. Treating cells with the mAb inhibited OMV uptake in a dose-dependent manner, with IC50 values of 3.85 µg/mL in A549 and 0.13 µg/mL in Caco-2 cells (Fig 2H). Using this mAb, we assessed the role of TIM-1 in OMV uptake in eleven additional human cell lines reported to express TIM-1 [33], representing diverse tissue types (769-P, ACHN, IGROV-1, Suit-2, Huh-7, SNU-449, 786-O, HCC1534, A-704, Cal-51, RPTEC). Antibody treatment reduced uptake in all cell lines by at least 40% (Fig A in S1 Text) including primary RPTEC cells. In contrast, cell lines lacking TIM-1 (FU97, HeLa, HCT-116, HT-29, THP-1) were largely unaffected by the anti-TIM-1 mAb. The observed variability in sensitivity to the TIM-1 mAb treatment in the TIM-1-positive cell lines could be a result of heterogeneity in TIM-1 surface levels or represent the presence of redundant OMV binding receptors. Altogether, we have identified TIM-1 as a significant and targetable driver of receptor-mediated OMV internalization.

## TIM-1 interacts with LPS in OMVs

The canonical TIM-1 ligand is phosphatidylserine (PS) [39]. However, this anionic phospholipid is only transiently present in *E. coli* making up ~0.1% of the detectable lipid pool [40], pointing to an alternative mechanism of interaction. Another known TIM-1 ligand that is present in OMVs is phosphatidylethanolamine (PE). While PE is predominantly contained within the inner leaflet of OMVs, we nonetheless tested whether it contributes to TIM-1 binding. Despite human TIM-1

(hTIM-1) and mouse TIM-1 (mTIM-1) being largely conserved, the human receptor is uniquely capable of binding PE [41]. We hypothesized that if PE mediates OMV binding, we should observe a reduction in affinity for mTIM-1 when compared to hTIM-1. After confirming PE specifically binds hTIM-1 and not mTIM-1 (Fig B in S1 Text), we tested this hypothesis. Both hTIM-1 (Fig 3A) and mTIM-1 (Fig 3B) demonstrated a strong dose dependent binding with OMVs, indicating that PE is unlikely to be driving the binding interaction.

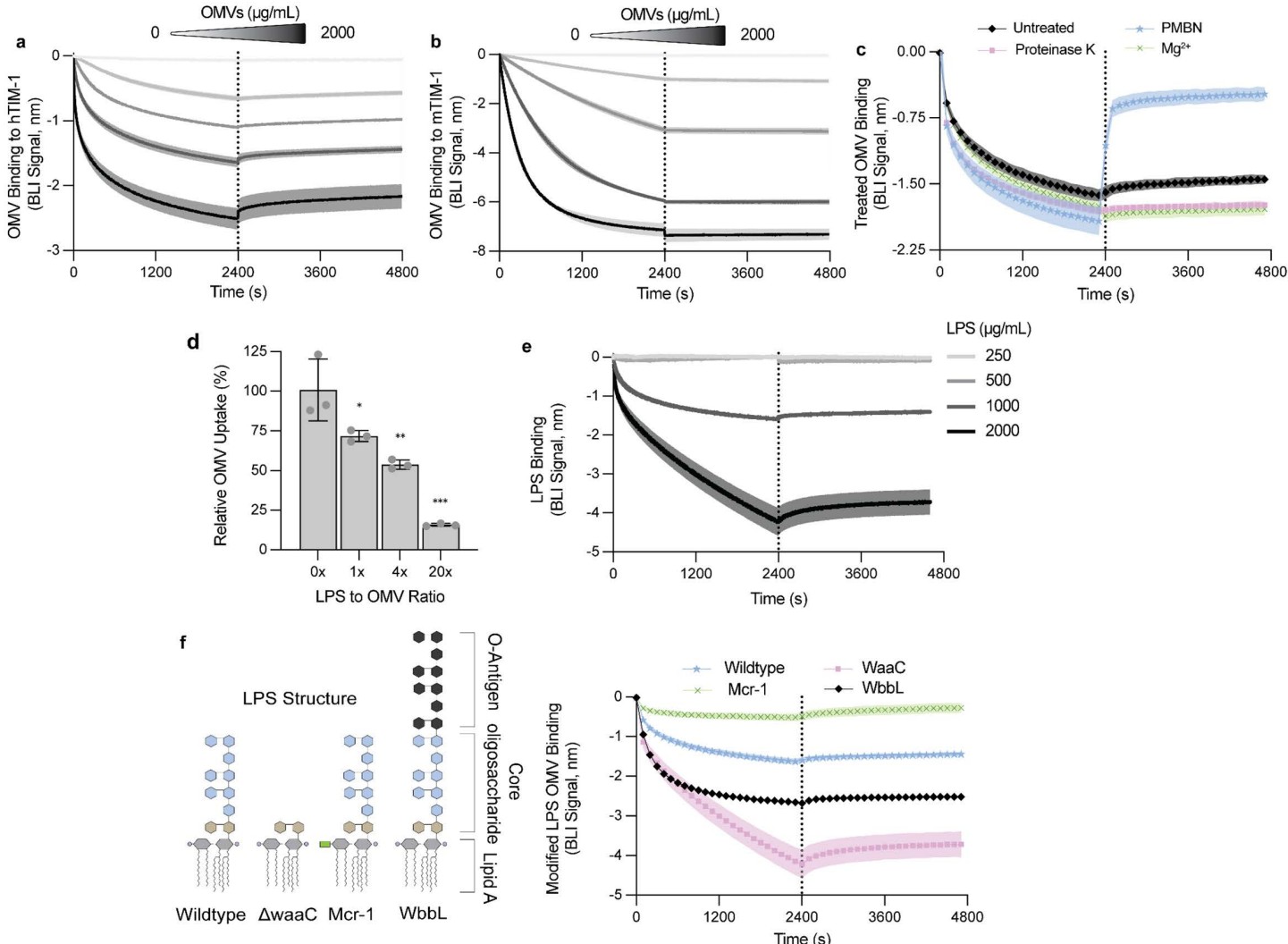

**Fig 3. TIM-1 recognizes the LPS on OMVs. (A–B)** Biolayer interferometry (BLI) of OMV binding to **(A)** human TIM-1 (hTIM-1) and **(B)** mouse TIM-1 (mTIM-1) across an OMV concentration series (2000, 500, 125, 31.25, 0 µg/mL). **(C)** BLI of OMVs treated as indicated to hTIM-1: proteinase K, 10 µg/mL polymyxin B nonapeptide (PMBN), or 10 mM $Mg^{2+}$. **(D)** A549 cells co-treated with DiO-labeled OMVs (50 µg/mL) and increasing concentrations of LPS from K12 *E. coli* (50, 200, 1000 µg/mL) for 2 hours. Uptake was measured by flow cytometry and normalized to an untreated control (= 100%). LPS to OMV mass ratios are shown on the x-axis. Bars represent mean±s.d. (n=3) *** $P < 0.001$, ** $P < 0.01$, * $P < 0.05$, One-way ANOVA (two-sided) with Dunnett's multiple comparisons vs 0x LPS. **(E)** BLI traces of nanoparticles generated from LPS binding to hTIM-1 at the indicated concentrations. **(F)** Schematic of LPS structure of genetically modified *E. coli* K-12 strains including lipid A (grey), Kdo (gold), core oligosaccharides (blue), and O-antigen (black). The green rectangle represents the addition of phosphoethanolamine by MCR-1 to lipid A. BLI of TIM-1 binding to OMVs from strains with altered LPS: wildtype, *waaC* mutant (core truncation), *wbbL* complementation (O-antigen restored), and *mcr-1* expression (phosphoethanolamine-modified lipid A. For BLI traces each solid line represents the mean and the shaded band is±s.d. of at least two biological replicates.

The OMV surface is predominantly composed of LPS, outer membrane proteins, lipoproteins, and polysaccharides [42]. To investigate which of these potential ligands facilitates the binding of TIM-1 to OMVs, we treated vesicles to modify the surface structure. Proteinase K treatment was used to remove surface proteins, and polymyxin B nonapeptide (PMBN) to bind the lipid A moiety of lipopolysaccharide (LPS) and reduce its anionic charge [43, 44]. Proteinase K treatment did not alter TIM-1 binding (Fig 3C). In contrast, PMBN reduced binding in the BLI assay by increasing the rate of dissociation (Fig 3C). Because PMBN is cationic and binds the phosphate groups of lipid A in LPS, the observed reduction in binding could reflect either the occlusion of LPS from TIM-1 or neutralization of the OMV surface charge. To specifically test the role of electrostatics, we measured OMV binding in the presence of excess $Mg^{2+}$ and found no effect (Fig 3C). This suggests the PMBN-dependent reduction in binding is not the result of reduced charge-based interactions and is instead consistent with PMBN blocking TIM-1 access to the lipid A moiety of LPS.

Having identified LPS as a potential TIM-1 ligand and therefore important for OMV internalization, we tested whether LPS can directly compete with OMVs for uptake. Increasing concentrations of LPS reduced OMV internalization in A549 cells (Fig 3D), suggesting LPS is a shared ligand for uptake. We next tested if TIM-1 can directly bind LPS in the BLI assay and detected LPS binding to TIM-1 at concentrations ≥1000 µg/mL (Fig 3E). LPS binding at these concentrations is not indiscriminate as we did not observe binding to another STM protein EGFR that served as a negative control (Fig C in S1 Text). We propose that as LPS micelle size increases with concentration (Table C in S1 Text), the observed threshold for detection could reflect a requirement for lipid nanoparticles to be of sufficient size to engage TIM-1 or represent a possible limitation in the sensitivity of our assay.

To isolate which specific component of LPS engages TIM-1, we altered LPS structure in our *E. coli* strain by genetic manipulation, isolated OMVs from these bacteria and tested for TIM-1 binding using BLI (Fig 3F). First, we generated OMVs from a *waaC* knockout, which lacks the heptose and galactose sugars of the LPS core. These OMVs retained strong TIM-1 binding, indicating that these sugars are not required for recognition (Fig 3F). We were also interested in testing if the presence of O-antigen would impede TIM-1 binding as O-antigen can facilitate OMV entry through non-receptor-mediated routes of endocytosis [29]. O-antigen production was restored in the *E. coli* K-12 strain by the expression of *wbbL,* as previously described [45], and these O-antigen-bearing OMVs maintained TIM-1 binding (Fig 3F). We next modified LPS by expressing the phosphoethanolamine transferase, mobile colistin resistance 1 (MCR-1), which adds phosphoethanolamine to the lipid A phosphates of LPS. OMVs from MCR-1–expressing cells showed reduced TIM-1 binding (Fig 3F) despite similar vesicle size (Table C in S1 Text), pointing to a dependence on the native lipid A structure for binding. Notably, OMVs from the LPS modified strains maintained their capacity for internalization in A549 and Caco-2 cells (Fig D in S1 Text) with ΔwaaC and WbbL showing increased uptake and OMVs from *E. coli* expressing *mcr-1* demonstrating impaired entry. Together, these results indicate that TIM-1 binds LPS and point to the lipid A moiety as the primary binding determinant mediating TIM-1-dependent OMV uptake.

## OMV uptake via TIM-1 requires an intact PS-binding pocket

TIM-1 is a type I membrane glycoprotein whose N-terminal IgV domain contains a metal-ion-dependent ligand-binding pocket that interacts with PS (Fig 4A). This binding pocket is defined by four signature residues, W112, F113, N114, and D115, which facilitate PS-binding (Fig 4B) and whose modification reduces TIM-1–dependent viral entry [46]. We tested whether OMVs might also be using this binding pocket for entry. In cell uptake assays, co-incubation with PS liposomes, but not phosphatidylcholine (PC) liposomes, significantly reduced OMV entry, suggesting competition for the PS-binding pocket of TIM-1 (Fig 4C). However, to specifically probe the importance of the WFND motif in OMV uptake TIM-1-knockout A549 cells were reconstituted with wildtype TIM-1 or modified TIM-1 in which this sequence was mutated to AAAA (Fig 4D). Wildtype TIM-1 restored OMV uptake above the vector control. However, despite similar surface levels of the mutant receptor (Table D in S1 Text), expression of TIM-1 (AAAA) did not significantly increase uptake, indicating a requirement for an intact PS pocket in TIM-1-mediated OMV entry.

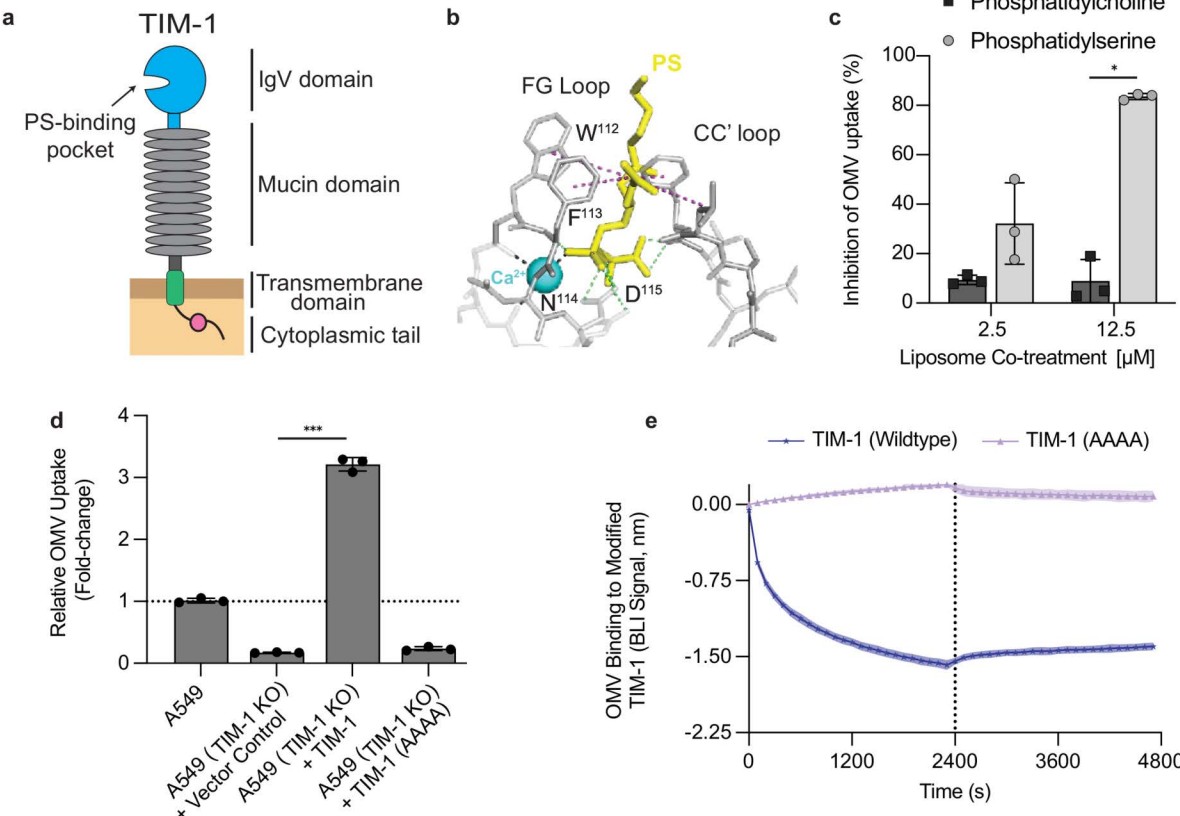

**Fig 4. TIM-1's phosphatidylserine (PS) pocket is required for OMV binding and uptake. (A)** Diagram of TIM-1 domain architecture highlighting the N-terminal IgV domain and PS-binding site. **(B)** Structural cartoon of the IgV pocket showing the WFND motif (W112/F113/N114/D115) and Ca²⁺ that coordinates phosphatidylserine (PS) binding. This structure is modeled using the PS-TIM-4 complex as a template (PDB: 3BIB) as previously described [41] and shows three major types of interactions formed between PS and hTIM-1 residues: Calcium-mediated interactions are shown by black dashed lines, hydrogen bonds by green dashed lines, and hydrophobic interactions by purple dashed lines. **(C)** Liposome competition during OMV uptake. A549 cells were exposed to DiO-labeled OMVs (50 µg/mL) for 2 hours with trypan blue quenching alongside PS or phosphatidylcholine (PC) liposomes at the indicated concentrations. Uptake normalized to no-liposome control (= 100%). Bars represent the mean ± s.d. (n = 3), * P < 0.05, two-sided unpaired t-test. **(D)** Requirement for the WFND pocket in OMV uptake. TIM-1-knockout A549 cells expressing empty vector, wildtype TIM-1, or WFND pocket mutant (AAAA) TIM-1 were exposed to DiO-labeled OMVs (50 µg/mL) for 2 hours. Uptake was normalized to wildtype A549 cells (=1.0). Bars represent mean ± s.d. (n = 3), *** P < 0.001, one-way ANOVA with Dunnett's multiple comparisons vs vector. **(E)** OMV binding to TIM-1 variant by biolayer interferometry (BLI). Solid line represents the mean and the surrounding band is ± s.d of at least 2 biological replicates.

To distinguish whether the impaired OMV uptake is the result of reduced OMV binding or impaired internalization signaling, we measured binding to the modified TIM-1 receptors using BLI (Fig 4E). The AAAA modification abolished OMV binding demonstrating that an intact PS-binding pocket mediates OMV binding and is crucial to facilitating OMV uptake.

## TIM-1 regulates OMV cytotoxicity

Internalization of OMVs delivers PAMPs into host cells and triggers proinflammatory cytokine secretion [4, 5]. THP-1 monocytes are particularly sensitive to OMVs, with cytosolic LPS triggering non-canonical inflammasome activation, inducing IL-1β, and pyroptotic cell death [13]. THP-1 monocytes did not produce detectable levels of TIM-1 [33], so we ectopically expressed this receptor to test whether OMV uptake through a TIM-1-dependent mechanism can elicit an inflammatory response. TIM-1 expression increased OMV uptake (Fig E in S1 Text) and, in turn, significantly increased cytotoxicity (Fig 5A) and IL-1β secretion in THP-1 cells (Fig 5B).

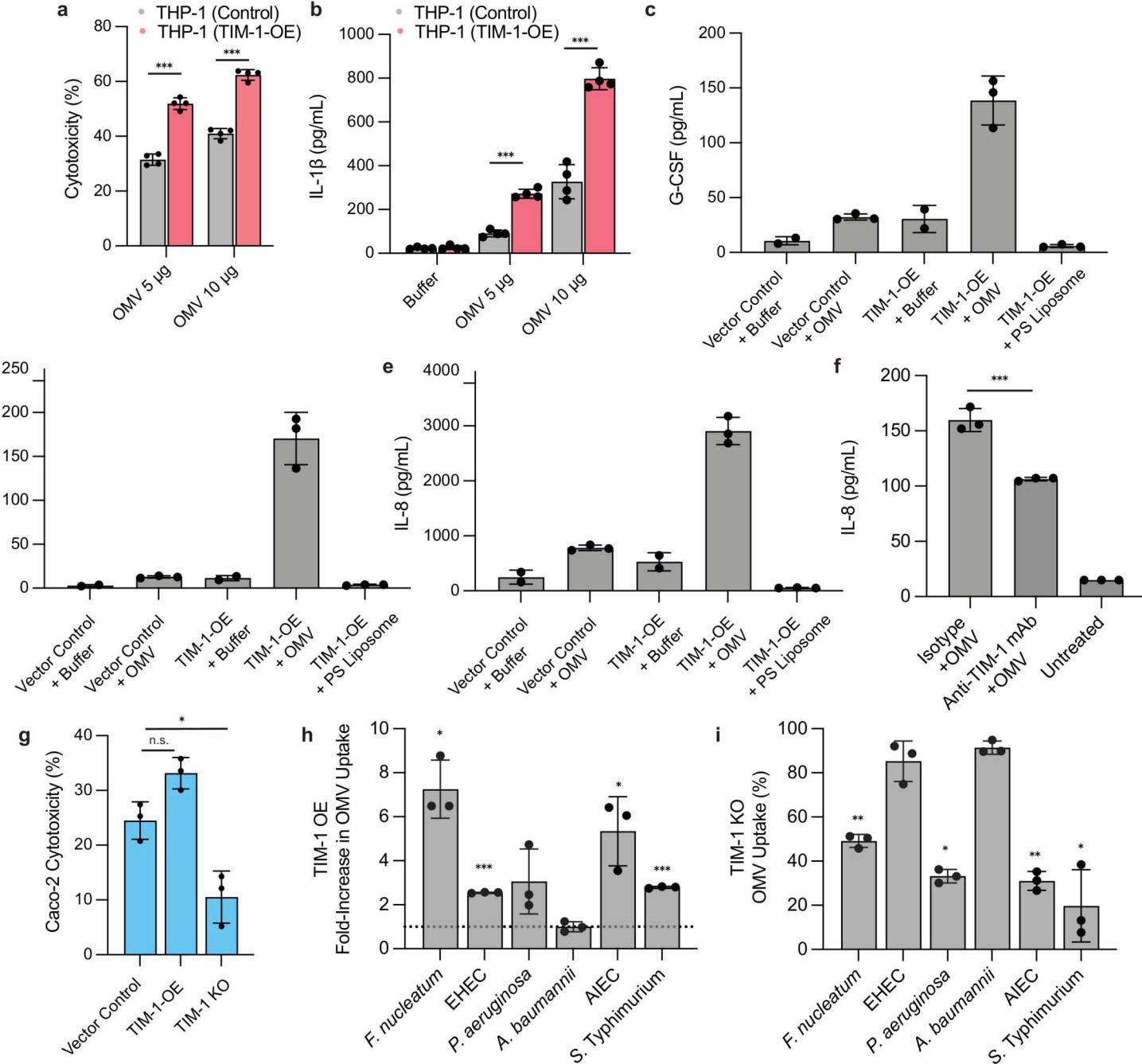

**Fig 5. TIM-1 modulates OMV-driven cytotoxicity, cytokine release, and uptake across cell types and bacterial species. (A)** THP-1 monocytes with or without TIM-1 overexpression (OE) were exposed to OMVs at the indicated doses. Cytotoxicity was quantified after 24 h**.** Bars represent the mean±s.d. (n=4) *** P<0.001, compared to the vector control, two-sided unpaired t-test. **(B)** IL-1β secretion from the cells in **(A)**. Bars represent the mean±s.d. (n=4) *** P<0.001, compared to the vector control, two-sided unpaired t-test. **(C–E)** TIM-1-OE A549 epithelial cells were treated as labeled, OMV and PS liposome concentration were at (200 μg/mL). Secreted G-CSF **(C)**, IL-6 **(D)**, and IL-8 **(E)** were measured after 24 hours of treatment. **(F)** IL-8 was quantified from A549 cells preincubated with anti-TIM-1 monoclonal antibody (mAb) or an isotype control before OMV (200 μg/mL) treatment. Bars represent the mean±s.d. (n=3) *** P<0.001, compared to the isotype control, two-sided unpaired t-test. **(G)** Caco-2 cells with empty vector, TIM-1-OE, or TIM-1 knockout (KO) were exposed to OMVs (200 μg/mL) and cytotoxicity measured after 24 hours. Bars represent the mean±s.d. (n=3) * P<0.05, n.s. (not significant) compared to the vector control, two-sided unpaired t-test. **(H)** TIM-1 overexpression in A549 cells alters uptake of OMVs from the multiple bacterial species. Values are fold-change relative to a vector control A549 cell line (= 1). **(I)** OMV uptake in TIM-1-knockout A549 cells

relative to a scrambled control for the species shown (=100). **(H)** and **(I)** Bars represent mean ± s.d, of at least three biological replicates *** P < 0.001, ** P < 0.01, * P < 0.05, using one-sample t-test vs 100% (two-sided).

OMVs can also induce robust cytokine production in epithelial cells [14,47,48]. We found secretion of G-CSF, IL-8, and IL-6 in A549 cells after OMV treatment (Fig 5C, 5D, and 5E). TIM-1 overexpression further amplified production of these cytokines (Fig 5C, 5D, and 5E) which was not simply a byproduct of increased TIM-1 internalization, as PS liposome treatment did not induce an inflammatory response. mAb blockade of TIM-1 in A549 cells reduced IL-8 secretion (Fig 5F). In Caco-2 cells, TIM-1 overexpression increased OMV-induced cytotoxicity, whereas TIM-1 knockout was sufficient to reduce cytotoxicity (Fig 5G).

Encouraged by the capacity of TIM-1 to maintain binding with OMVs containing intact O-antigen (Fig 3F), we assessed whether this receptor facilitates entry of OMVs from bacterial species beyond the laboratory *E. coli* K-12 strain. TIM-1 overexpression increased uptake of OMVs from *F. nucleatum*, *P. aeruginosa*, enterohemorrhagic *E. coli* (EHEC), adherent-invasive *E. coli* (AIEC), and *Salmonella* Typhimurium, but not *Acinetobacter baumannii* (Fig 5H). However, increased uptake with TIM-1 overexpression did not necessarily correlate with a reduction in the TIM-1-knockout background. Uptake of *F. nucleatum*, AIEC, *S.* Typhimurium, and *P. aeruginosa* OMVs was reduced in TIM-1-knockout A549 cells, whereas uptake of EHEC OMVs was not impaired (Fig 5I). These results suggest that TIM-1 can function as a primary entry receptor for some OMVs and may act as a facilitator that augments redundant or additional OMV-entry receptors that are yet to be identified. Altogether, our data support TIM-1 as a bona fide receptor for OMVs, and a potential druggable target to reduce OMV uptake and dampen downstream OMV-mediated pathogenesis.

## Discussion

To identify receptors that bind OMVs, we screened a library of human single-pass transmembrane (STM) proteins. Hits were enriched for receptors that recognize anionic polysaccharides, which is consistent with the negative surface charge of OMVs. In follow-up binding and uptake assays, the phosphatidylserine-binding receptors TIM-4 and TIM-1 facilitated OMV internalization. Other PS-binding receptors that engaged OMVs were identified as binders in the STM screen but were not prioritized for follow-up, including CD300A, CD300LG, and CD300C (Table A in S1 Text). PS-binding capacity did not always predict OMV binding, as stabilin-1, stabilin-2, and TIM-3, which were present in the STM screen, were not identified as OMV binders. However, further investigation of these PS-binding receptors is warranted as the Fc-chimera structure used for screening can impair protein function.

Disrupting TIM-1 by knockout or with a blocking antibody markedly reduced OMV uptake across diverse cell lines. Notably, while we utilized the lipophilic dye DiO with trypan blue quenching to monitor internalized OMVs, lipophilic labeling can be limited by dye transfer to host membranes or potential alterations to vesicle surface structure [26]. Consideration should also be given for the potential impact OMV purification methods may play in receptor binding affinity. However, the close agreement between our BLI binding data, cellular uptake, and functional response assays collectively supports TIM-1-dependent vesicle internalization.

We were unable to completely abolish OMV uptake, suggesting that other surface receptors contribute to entry, or a low level of entry occurs through receptor-independent pathways such as membrane fusion. Additionally, TIM-1-negative cell lines were found to internalize OMVs, indicating that more receptors remain to be elucidated. Ongoing efforts are focusing on characterizing the role of the remaining OMV-binding receptors uncovered in the STM screen in both TIM-1-positive and TIM-1-negative backgrounds. This work was restricted to STMs that perform well in our screening platform and does not interrogate multipass TM proteins that might also serve as OMV receptors. We focused our efforts on elucidating the mechanism of binding between OMVs and TIM-1. OMV binding and cell uptake assays identified an interaction between TIM-1 and LPS that relies on an intact phosphatidylserine binding pocket. PMBN was used to mask the lipid A moiety of

LPS, which is known to bind broadly to anionic phosphate groups, including those on phosphatidylserine (PS). While our *E. coli* OMVs contain negligible amounts of PS, the fact that PMBN disrupts TIM-1 binding to these vesicles suggests a shared biophysical requirement for phosphate recognition. This is consistent with our finding that OMV entry is competitively inhibited by PS liposomes and requires an intact WFND pocket. Rather than inducing a non-specific conformational change in the receptor, PMBN likely disrupts the electrostatic interaction between the TIM-1 binding pocket and the anionic phosphate headgroups common to both PS and lipid A.

Nanoparticles generated from pure LPS also bound TIM-1, but only at concentrations that generate micelles larger than roughly 100 nm. This size dependence may reflect detection limits of the binding assay. Alternatively, TIM-1 could require its target to be presented in vesicles above a threshold size to support binding and uptake. Such a requirement would be consistent with reported size biases in OMV entry routes, with smaller vesicles often using lipid-raft–dependent pathways, mid-sized vesicles favoring clathrin-mediated endocytosis, and larger vesicles relying on macropinocytosis [28].

LPS structure varies across Gram-negative species, so we tested whether TIM-1 interacts with OMVs from bacteria beyond our laboratory *E. coli* strain. As O-antigen has been shown to play an important role in mediating OMV entry kinetics [29], we first tested the impact of restoring O-antigen in our laboratory *E. coli* strain on TIM-1 binding. As these OMVs were still capable of binding TIM-1, we investigated bacteria whose OMVs have been shown to contribute to pathogenesis during infection. Overexpression of TIM-1 increased uptake of OMVs from *F. nucleatum*, *P. aeruginosa, S.* Typhimurium, AIEC, and EHEC, but not from *A. baumannii*. *A. baumannii* lipid A is hepta-acylated, compared to the hexa-acylated structure of *E. coli*. *A. baumannii* also produces a unique lipooligosaccharide and thick capsular polysaccharide, which could limit TIM-1 access. Notably, OMVs from this bacterium are reported to enter cells mainly through lipid-raft–dependent pathways [49], consistent with TIM-1-independent uptake observed in this study. TIM-1 knockout had little effect on EHEC, but reduced uptake for *F. nucleatum*, AIEC, *S.* Typhimurium, and *P. aeruginosa.* EHEC could be utilizing another receptor for internalization that can fully compensate for the loss of TIM-1. Altogether these data indicate that TIM-1 can act as the dominant entry receptor for some OMVs and as a co-receptor that augments alternative uptake routes for others.

Inhibiting OMV entry has been proposed as a novel therapeutic approach to reduce bacterial pathogenesis [4, 50], however, this has been difficult to validate. The limited genetic control over OMV biogenesis has prevented selective shutdown of vesicle production in bacteria, so most studies have relied on OMV-only treatments to demonstrate that vesicles are sufficient for pathology and likely contribute to infection outcomes. On the host side, the incomplete understanding of OMV entry and cellular trafficking has also prevented testing of the importance of OMV internalization for pathogenesis. Broad endocytosis inhibitors confound interpretation because they have off-target effects such as blocking internalization of Toll-like receptors, making it hard to separate surface signaling from responses to intracellular PAMPs.

Identifying TIM-1 has allowed us to directly link a defined receptor to OMV uptake and downstream inflammation. Entry through TIM-1 elicits proinflammatory cytokine release, and anti-TIM-1 mAb treatment reduces both uptake and OMV-driven pathology, indicating that TIM-1 is druggable. Reliance on TIM-1 by pathogens such as *P. aeruginosa*, AIEC, *S.* Typhimurium, and *F. nucleatum* supports future in vivo testing. Growing evidence implicates *F. nucleatum* in colorectal and other cancers [51] which are known to overexpress TIM-1 [52]. This study suggests a potential mechanism by which Fusobacterium-derived OMVs interact with host cancer cells. This interaction requires further investigation to fully understand its implications. Notably, reduction of TIM-1 in mice shows reduced sensitivity to LPS-induced sepsis [53], previously attributed to altered immune signaling. Our data raise the possibility that diminished cellular internalization of LPS may also contribute to this protection. Evaluating the efficacy of targeting TIM-1 in vivo for its ability to curb OMV-mediated disease will be an important next step in validating this OMV-focused antivirulence approach.

OMV biodistribution is an important consideration for OMV-based delivery technologies. TIM-1 has been found to be upregulated in several tumor types [52] including renal cell carcinoma, hepatocellular carcinoma, lung cancer, and colon cancer which might be leveraged for tumor-directed OMV delivery. It is interesting to consider that while TIM-1 expression is typically at low or undetectable levels in healthy human tissues, its reported induction during cell stress, including

extracellular LPS through TLR4-mediated signaling [53] provides a pathway for OMV-induced receptor upregulation. This suggests a potential mechanism whereby the presence of OMVs could actively facilitate their own cellular uptake by stimulating the expression of TIM-1. Surface display of targeting ligands can also be used to improve tissue-specific delivery of therapeutic cargo, but substantial off-target uptake still occurs [54,55]. Because we have identified TIM-1 as recognizing LPS on the OMV surface, rational edits to the lipid A moiety that diminish TIM-1 recognition could be combined with targeting ligands to improve on-target OMV delivery. Future in vivo studies should evaluate the role of TIM-1 in OMV distribution by administering vesicles in the presence of a TIM-1 blocker.

In this work, we asked how OMVs enter host cells and whether a defined receptor initiates that process. Using a human receptor screen and follow-up assays, we identified TIM-1's ability to bind OMVs and mediate OMV entry. Mechanistic studies show that LPS found on OMVs interacts with the PS-binding pocket of TIM-1. Targeting TIM-1 can reduce OMV uptake and downstream pathogenesis. These results move the field from broad pathway inference to a receptor-level mechanism and point to a tractable, host-directed strategy to limit vesicle-mediated pathogenesis while informing efforts to engineer OMVs for vaccines and drug delivery.

## Materials and methods

### Bacterial strains and culture conditions

Bacterial strains and their sources are described in Table E in S1 Text. Unless otherwise noted, bacterial cultures were grown by picking a single bacterial colony from Luria-Bertani (LB) agar plates into LB broth medium and grown overnight at 37°C. Kanamycin (25 µg/mL) and carbenicillin (50 µg/mL) were used when antibiotic selection was required.

### Cell culture

Cell lines and their sources are listed in Table F in S1 Text. Cells were cultured in either Dulbecco's Modified Eagle Medium (DMEM) containing 10% (v/v) fetal bovine serum (FBS) and 2 mM glutamine or RPMI containing 10% (v/v) FBS, 2 mM glutamine, and 1 mM sodium pyruvate or DMEM:F12 RPTEC complete growth media (ATCC). Cells were incubated at 37°C in a 5% $CO_2$ humidified incubator, passaged two to three times a week, and used for experiments at passages between 8 and 25.

### OMV preparation and labeling

OMVs were isolated as previously described [56]. Briefly, bacteria were grown in LB overnight at 37°C with aeration. Cells were pelleted by centrifugation at 10,000 × g for 30 min at 4°C. Supernatants were filtered through a 0.45 µm PVDF filter (Sigma Aldrich) and concentrated via tangential flow filtration (Vivaflow 200, MWCO 100kDa). OMVs were pelleted by ultracentrifugation at 185,000 × g for 2 hours at 4°C. OMV pellets were washed in 50 ml of OMV buffer (phosphate buffered saline with 200 mM NaCl, 1 mM $CaCl_2$, and 0.5 mM $MgCl_2$), resuspended and ultracentrifuged again at 185,000 × g for 2 hours. The OMV pellet was resuspended in OMV buffer, concentrated, and washed three times in a 100kDa MWCO filter (Amicon) and finally passed through a 0.45 µm PVDF syringe filter (Millex). OMV preparations were quantified using a Pierce BCA protein assay kit (Thermo Fisher). Aliquots were stored at 4°C or −80°C. *E. coli* OMVs were isolated from the hypervesiculating ΔtolQ strain in all experiments except for the generation of ΔwaaC and NanoLuc containing OMVs which used the wildtype background.

OMV concentration was determined using BCA (Thermo) and nanoparticle tracking analysis. OMVs were fluorescently stained as previously described [30]. Samples were labeled with Vybrant DiO cell-labeling solution (Invitrogen) by incubating at 37°C for 30 min at 2.5% v/v. Excess dye was removed by washing four times with PBS using a 100 kDa MWCO filter unit (Amicon).

## OMV uptake assays

The day before the assay cells were seeded in 24 well plates at a density of 0.75–1.5 × 10$^5$ cells per well. Cells were washed twice with PBS followed by addition of 300 μL of reduced serum Opti-MEM (Gibco) media and treated with DiO-labeled OMVs at 50 μg/mL (1.98 x 10$^{11}$ particles per mL), unless mentioned otherwise for 2 hours. Uptake assays were conducted with an incubation time of 2 hours as >95% of the population is OMV positive at this timepoint and uptake has not saturated. After incubation cells were washed with PBS, detached with TrypLE (Gibco), collected and washed with FACS buffer (PBS, 0.5% BSA, 0.05% sodium azide), resuspended in 200 μL of FACS buffer and 50 μL of 0.4% trypan blue, kept on ice, and fluorescence quantified by flow cytometry. Cells were analyzed using a BD FACSymphony (HTS) flow cytometer, and data processed using FlowJo analysis software.

For chemical inhibition studies, cells were pre-treated with 15 μM filipin III (Sigma-Aldrich), 100 μM dynasore (Cayman Chemical), 20 μM blebbistatin (Fisher Scientific), or 42 μM chlorpromazine (Cayman Chemical) for 1 hour. Uptake assays were then performed as described above with inhibitors maintained in the media. For antibody blockade, cells were pre-treated with either anti-human CD365 (TIM-1) mAb (1D12, BioLegend), or mouse IgG1, κ isotype Ctrl antibody (BioLegend) for 30 minutes before addition of OMVs. Maintaining the antibody in the media, uptake assays were performed as described above.

OMV uptake competition assays were conducted by first treating cells with LPS-EK from *E. coli* K-12 (InvivoGen), phosphatidylserine liposomes (CPS-603, Cellsome), or phosphatidylcholine liposomes (CPC-606, Cellsome). Immediately after cells were treated with OMVs and the uptake assays performed as described above.

## Receptor overexpression

The PiggyBac transposon system was used to stably integrate receptors of interest into A549, THP-1, and Caco-2 cell lines. The PiggyBac transposon vectors each contained the receptor gene of interest under the control of the EF-1α promoter, alongside an mScarlet fluorescent reporter gene for visualization and IRES-Puromycin for selection. The PiggyBac transposase expression plasmid (pCMV-PBase) was co-transfected with the transposon plasmid (3:1 transposon to transposase ratio) to facilitate genomic integration. Transfection was performed using the Lonza 4D-Nucleofector system according to the manufacturer's recommended protocol for each cell line. After transfection, cells were immediately transferred to pre-warmed DMEM or 20% FBS RPMI (THP-1 cells) and incubated at 37°C in a 5% CO$_2$ humidified incubator. Five days post-transfection, transformants were treated with 2 μg/mL puromycin (A549 cells), 10 μg/mL puromycin (Caco-2 cells), or 4 μg/mL puromycin (THP-1) for 7 days to enrich for stably transfected cells. Cells were then maintained at 2 μg/mL puromycin. Receptor expression and successful integration were confirmed by flow cytometry using the mScarlet fluorescent reporter. Surface receptor expression for TIM-1 was assessed using the Alexa Fluor 488 conjugated anti-TIM-1 antibody (P365D, Invitrogen), TIM-4 using APC conjugated anti-TIM-4 (9F4, BioLegend), and NRP1 using FITC conjugated anti-NRP-1 (12C2, BioLegend) using a standard staining protocol and flow cytometry as described above (Table D in S1 Text). Notably WBP1 overexpression and Caco-2 cell line surface expression was not confirmed.

## CRISPR-Cas9-mediated gene knockout

Knockout of *TIM-1* in A549 and Caco-2 cells was performed using the CRISPR-Cas9 system. Single-guide RNA (sgRNA) targeting *TIM-1* (also known as HAVCR1) was designed using the crRNA sequence Hs.Cas9.HAVCR1.1.AE (Integrated DNA Technologies, IDT). The gRNA duplex was prepared by combining crRNA with tracrRNA (IDT), heating to 95°C for 5 minutes, and cooling to room temperature. The ribonucleoprotein (RNP) complex was assembled by mixing 1.2 pmol gRNA, 1 pmol Alt-R S.p. Cas9 Nuclease V3 (IDT), and 1 pmol electroporation enhancer (IDT), followed by incubation at room temperature for 30 minutes. Transfection of complexes was performed using the Lonza 4D-Nucleofector system

according to the manufacturer's recommended protocol for each cell line. Five days post-nucleofection, cells were stained for TIM-1 as described above and sorted for the absence of TIM-1 using a BD Fusion/S6 cell sorter.

## Biolayer interferometry

Biolayer interferometry (BLI) analysis of OMVs, LPS, PS (ASTATECH, H12024), PC (Sigma-Aldrich, P3556) was adapted from previous BLI efforts using eukaryotic vesicles [31,32]. In short, BLI was performed on an Octet RED96e using Fc-capture (AHC) (Sartorius) or His-tag biosensors (Sartorius). Sensors were hydrated for 5–10 min at room temperature in the OMV buffer. rhTIM-1-Fc (R&D Systems, 9319-TM), rhTIM-4-Fc (R&D Systems, 9300-TM), rhNRP1-Fc (R&D Systems, 10455-N1), rhEGFR-Fc (R&D Systems, 344-ER), rmTIM-1-Fc (R&D Systems, 10699-TM), rhWBP1-His (Bon Opus, CE39), and rhAPLP2-His (Sino Biological, 30157-H08H), chimeric proteins were diluted in OMV buffer to 50 nM. Lipids were diluted in OMV buffer. All solutions (≥150 µl per well) were dispensed into a black polypropylene chimney 96-well plate (Greiner 655900). Assays were run at 30°C with 1000 rpm orbital shaking using the following sequence: baseline in OMV buffer for 240 s, ligand loading for 600 s (Receptor-Fc/His), baseline in OMV buffer for 240 s, association in ligand for 2400 s, and dissociation in OMV buffer for 2400 s. Reference sensors in buffer were included for subtraction, and inter-step correction was applied when appropriate. Lipid vesicles produced the expected negative response due to their large size as previously observed [31,32]. Modified TIM-1 PS-binding pocket mutant-Fc protein was expressed and purified in house from ExpressCHO cells and verified using mass spectroscopy.

OMVs were treated to assess impact on TIM-1 binding. 100 µL of OMVs at a concentration of 2 mg/mL were treated with 2 µL of 120 units/mL thermolabile proteinase K (NEB), incubated at 37°C for 2 hours followed by 55°C for 15 minutes to inactivate the enzyme and purified using a 100 kDa MWCO filter unit (Amicon). Polymyxin B nonapeptide (Sigma Aldrich) and 10 mM $MgCl_2$ were added directly to the 96-well plate in the association and dissociation wells during BLI. Direct interaction with pure LPS nanoparticles was conducted using LPS-EK from *E. coli* K-12 (InvivoGen).

## Receptor screen

OMVs from *E. coli* expressing ssDsbA:NanoLuc were diluted to a final concentration of 0.03 to 0.05 mg/mL (as measured by BCA) in OMV buffer + 1% bovine serum albumin Fraction V (Sigma Aldrich). Preparation of the human receptor library was performed using a robotic system consisting of liquid handling devices to allow for high-throughput analysis. To generate this library, Fc-tagged protein ectodomains were individually transfected in cells for expression and secretion as soluble proteins into the growth media. The conditioned media was then transferred to protein A–coated plates, with each well receiving a different ectodomain. This resulted in a collection of immobilized ectodomains suitable for high-throughput screening on white 384-well plates (Thermo Fisher Scientific). Plates were washed three times with an OMV buffer to remove unbound components. Vesicles were added to the plates and allowed to sit overnight at 4°C. To prevent clogging, 25 µL of OMV buffer was added to the plates. For a positive control used for normalization, 25 µL of the same vesicle stocks used in the screens were added into the first column of each plate after all washing steps were completed. A total of 25 µL 1 µM coelenterazine h (Promega) in OMV buffer was dispensed into the wells, incubated for 5 min, and read on a TECAN using 0.1 s of luminescence read time. Receptors with a normalized luminescence intensity ≥0.01 (approximately 2 standard deviations from the median), in both screening replicates were deemed as OMV binders. A detailed description of the screening platform and methodology has been previously described [31,32].

## Cytotoxicity and cytokine secretion

96-well plates were seeded with $10^4$ cells per well, incubated overnight, washed with PBS and grown in reduced serum Opti-MEM media. OMV cytotoxicity was assessed using CellTiter-Glo (Promega) after treatment of cells with OMVs for 18–24 hours. Broad cytokine secretion analysis was determined from supernatants using a multiplex Luminex assay.

Briefly, supernatants collected from A549 cells were analyzed for concentrations of 33 human cytokines using Milliplex MAP reagents (MilliporeSigma, St. Louis, MO) according to manufacturer's recommended protocol. Dilution of reconstituted standards were made at a 2.73-fold, to increase the number of data points from six to nine while maintaining original dynamic range. Fluorescence intensity was measured by xPONENT software v 4.2 on FlexMap 3D instruments (Luminex Corp, Austin, TX). Bio-Plex Manager v 6.2 (Bio-Rad Laboratories, Hercules, CA) was used to construct standard curves for each analyte on each plate using the median FI of at least 20 beads done in duplicate. A 4 or 5- point regression analysis was used to calculate best fit and cytokine levels. IL-1β and IL-8 were also quantified after 18 hours of OMV treatment using the Lumit IL-1β Human Immunoassay (Promega) for IL-1β and the Human IL-8 ELISA Kit (R&D Systems) following the standard protocols.

## Supporting information

**S1 Text. Table A. OMV-binding proteins.** Table B. GO molecular function enrichment for OMV-binding receptors. Table C. OMV size. Table D. Surface validation of receptor overexpression cell lines. Table E. Bacterial strains used in the study. Table F. Cell lines used for this study. Fig A. Anti–TIM-1 antibody reduces OMV uptake across TIM-1–expressing cell lines. Fig B. Human TIM-1 (hTIM-1) interacts with phosphatidylethanolamine (PE). Fig C. LPS does not indiscriminately bind receptors. Fig D. OMVs from LPS modified *E. coli* strains are internalized by A549 and Caco-2 cells. Fig E. Overexpression of TIM-1 in THP-1 monocytes.
(DOCX)

## Acknowledgments

We acknowledge Jillian Pattison for assistance with pBac vector design, Ryan Fong for TIM-1 mutant vector design and purification support, Steven Rutherford for intellectual support and manuscript review. We acknowledge the fruitful intellectual discussions with Dennis Wolan, Rick Brown, Sharookh Kapadia, Angel Jimenez, and Lindsey Carfrae.

## Author contributions

**Conceptualization:** Craig R. MacNair, Varnesh Tiku, Man-Wah Tan.

**Funding acquisition:** Man-Wah Tan.

**Investigation:** Craig R. MacNair, Varnesh Tiku, Shengya Cao, Ariana D. Sanchez, Barath Udayasuryan, Katharina Theresa Kroll, Adarsh Singh.

**Methodology:** Craig R. MacNair, Shengya Cao.

**Project administration:** Craig R. MacNair.

**Supervision:** Man-Wah Tan.

**Writing – original draft:** Craig R. MacNair.

**Writing – review & editing:** Craig R. MacNair, Varnesh Tiku, Shengya Cao, Ariana D. Sanchez, Barath Udayasuryan, Katharina Theresa Kroll, Adarsh Singh, Man-Wah Tan.

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
