## [Decision Letter · Decision Letter 0]

17 Mar 2026

PPATHOGENS-D-26-00125

Outer Membrane Vesicles Hijack TIM-1 for Cellular Uptake

PLOS Pathogens

Dear Dr. MacNair,

Thank you for submitting your manuscript to PLOS Pathogens. After careful consideration, we feel that it has merit but does not fully meet PLOS Pathogens's publication criteria as it currently stands. Therefore, we invite you to submit a revised version of the manuscript that addresses the points raised during the review process.

We look forward to receiving your revised manuscript.

Kind regards,

Eric Oswald, Ph.D., D.V.M.

Academic Editor

PLOS Pathogens

Thomas Guillard

Section Editor

PLOS Pathogens

Sumita Bhaduri-McIntosh

Editor-in-Chief

PLOS Pathogens

orcid.org/0000-0003-2946-9497

Michael Malim

Editor-in-Chief

PLOS Pathogens

orcid.org/0000-0002-7699-2064

**Journal Requirements:**

At this stage, the following Authors/Authors require contributions: Craig MacNair, Varnesh Tiku, Shengya Cao, Ariana D. Sanchez, Barath Udayasuryan, Katharina Theresa Kroll, Adarsh Singh, and Man-Wah Tan. Please ensure that the full contributions of each author are acknowledged in the "Add/Edit/Remove Authors" section of our submission form.

https://journals.plos.org/plospathogens/s/submission-guidelines#loc-parts-of-a-submission

Potential Copyright Issues:

i) Figures 1B, and 2A. Please confirm whether you drew the images / clip-art within the figure panels by hand. If you did not draw the images, please provide (a) a link to the source of the images or icons and their license / terms of use; or (b) written permission from the copyright holder to publish the images or icons under our CC BY 4.0 license. Alternatively, you may replace the images with open source alternatives. See these open source resources you may use to replace images / clip-art:

6) Please provide a detailed Financial Disclosure statement. This is published with the article. It must therefore be completed in full sentences and contain the exact wording you wish to be published.

1) Please clarify all sources of financial support for your study. List the grants, grant numbers, and organizations that funded your study, including funding received from your institution. Please note that suppliers of material support, including research materials, should be recognized in the Acknowledgements section rather than in the Financial Disclosure

2) State the initials, alongside each funding source, of each author to receive each grant. For example: "This work was supported by the National Institutes of Health (####### to AM; ###### to CJ) and the National Science Foundation (###### to AM)."

3) State what role the funders took in the study. If the funders had no role in your study, please state: "The funders had no role in study design, data collection and analysis, decision to publish, or preparation of the manuscript."

4) If any authors received a salary from any of your funders, please state which authors and which funders..

7) Your current Financial Disclosure states, "The author(s) received no specific funding for this work.".

However, your funding information on the submission form indicates recieving fund .

Please indicate by return email the full and correct funding information for your study and confirm the order in which funding contributions should appear. Please be sure to indicate whether the funders played any role in the study design, data collection and analysis, decision to publish, or preparation of the manuscript.

8) Please send a completed 'Competing Interests' statement, including any COIs declared by your co-authors. If you have no competing interests to declare, please state "The authors have declared that no competing interests exist". Otherwise please declare all competing interests beginning with the statement "I have read the journal's policy and the authors of this manuscript have the following competing interests"

**Reviewers' Comments:**

Reviewer's Responses to Questions

**Part I - Summary**

Reviewer #1: The study by MacNair et al. investigated the role of the TIM-1 receptor in the uptake of Escherichia coli's outer membrane vesicles (OMVs) by eukaryotic cells through clathrin-dependent endocytosis. Using mainly two cell lines (A549 and Caco-2), the authors demonstrated that TIM-1 binds OMVs via lipopolysaccharide (LPS), most likely through the lipid A moiety, which interacts with the phosphatidyl serine binding pocket of TIM-1. Notably, the authors showed the critical role of TIM-1 as entry of OMV is markedly impaired in TIM-1 knockout cells or in presence of TIM-1 blocking antibody in A549 and Caco-2 cell lines whereas other cell types display only partial or no inhibition of OMV uptake. Consistently, TIM-1 dependent delivery of OMV-associated PAMPs triggers downstream immune responses. The study further extends these observations to OMVs derived from other Gram-negative bacteria, including Pseudomonas and Fusobacterium, while uptake of OMVs from Acinetobacter baumannii and EHEC is not reduced in TIM-1 knockout cells.

Although TIM-1 dependency appears limited to a subset of cell lines (without involvement of this receptor observed in canonical cell lines such as HeLa, HCT116, HT29, or THP-1) and to OMVs derived from certain bacterial strains, this study is methodologically sound and identifies a novel receptor contributing to OMV endocytosis. Given the scarcity of characterized OMV receptors, although additional validation will be required, this study may represent a significant contribution to the field.

Reviewer #2: The manuscript titled “Outer Membrane Vesicles Hijack TIM-1 for Cellular Uptake” demonstrates the involvement of endocytic pathways in OMVs uptake using various blockers. Although there are similar studies in the past (ex. Khan et al., 2024, ACS Inf Dis), the present work covers different types of bacteria. This work further advances existing knowledge by demonstrating that TIM-1 is a strong binding receptor for OMVs uptake across different cell types (A549, Caco-2, and THP1) using gain and loss-of-function approaches. This study reported that TIM-1 binds LPS on the OMV surface via its phosphatidylserine binding domain and mediates receptor-mediated endocytosis. The manuscript is mostly well-written and well-presented.

However, the following comments to further improvement of the manuscript:

1. For the cellular uptake studies, the authors labelled OMVs with a lipophilic dye for flow cytometry analysis. The authors should briefly discuss potential limitations of lipophilic dyes in the Discussion section to better contextualize the uptake process.

2. Mention the rationale for choosing the time point (2 hours) for the uptake study.

3. As the authors have already established luciferase expression in E. coli, they should consider performing cellular uptake studies using luciferase-containing OMVs to evaluate OMV internalization. In particular, measuring luminescence would provide an complementary approach to assess TIM-1-mediated OMV uptake.

4. The purity of the OMVs should be carefully assessed. Since only ultracentrifugation (UC) was used for isolation, which may result in crude OMVs mixed with bacterial remnants or medium components, additional purification steps, such as size-exclusion chromatography (SEC) or density gradient centrifugation, are recommended, particularly when receptor-binding interactions are being investigated. This limitation should be briefly acknowledged in the Discussion.

5. The authors suggest blocking TIM-1 using PMBN due to its LPS-binding activity. However, since PMBN binds phosphate groups, it is unclear how this might affect TIM-1 interaction with PS. Possible conformation changes? Should discuss the potential impact of PMBN on TIM-1 binding to PS.

6. For studies involving the uptake of OMVs from different bacterial strains (5h- P. aeruginosa; 5i- P. aeruginosa and A. baumannii), the authors should include at least 3 biological replicates for reproducibility.

7. The authors may consider modifying the title of the manuscript, as the data indicate that TIM-1-mediated OMV uptake is strain specific. For example, no uptake of A. baumannii OMVs was observed, and only a ~2-fold increase was seen for E. coli OMVs in TIM-1-overexpressing A549 cells, while only no reduction in uptake for TIM-1 knockout cells.

8. The in vitro results demonstrate that TIM-1 serves as a receptor for E. coli K-12 OMV uptake; however, in vivo systems are more complex. To further evaluate the potential of TIM-1-mediated OMV uptake, the authors should discuss the possibility for a biodistribution study of OMVs in the presence of TIM-1-blocker.

Reviewer #3: In this manuscript, MacNair et al. report the identification of T-cell immunoglobulin and mucin-domain 1 (TIM-1) as a major host receptor mediating uptake of bacterial outer membrane vesicles (OMVs). Using a large-scale screen of over 1,500 human receptor proteins, the authors identify TIM-1 as a strong OMV-binding receptor. They demonstrate that TIM-1 overexpression enhances OMV internalization, whereas TIM-1 knockout and antibody blockade reduce uptake across multiple epithelial cell lines. Mechanistically, they propose that TIM-1 binds lipopolysaccharide (LPS) on the OMV surface via its phosphatidylserine-binding domain and that TIM-1-dependent uptake promotes proinflammatory cytokine production. Finally, they show that OMVs from multiple Gram-negative species exploit TIM-1 for entry.

OMVs are widely recognized as key mediators of host–microbe interactions and are increasingly explored as vaccine and drug delivery platforms. Although both clathrin-dependent and clathrin-independent pathways have been implicated in OMV uptake, specific host receptors remain poorly defined. The identification of a specific receptor therefore represents an important and timely conceptual advance.

**Part II – Major Issues: Key Experiments Required for Acceptance**

Reviewer #1: 1- The role of TIM-1 is investigated exclusively in transformed cell lines, which may represent a phenotype-related bias. Validation of the role of TIM-1, using at least blocking antibody, in non-transformed cell lines (such as HUVEC, IEC-6 etc...) is essential.

2- Using genetically engineered E. coli, the authors have investigated the role of LPS modifications to TIM-1 binding using biolayer interferometry assays. However, uptake of OMVs produced by such E. coli strains (∆wacc, WbbL, mcr-1) have to be evaluated in cellulo, at least in A549 and Caco-2 cells.

3- The fact that EHEC's OMV uptake is not impaired in TIM-1 knockout cell line suggests the existence of alternative TIM-1-independent entry routes. This raises the question of how representative OMVs from the model strain E. coli BW25113 are. To exclude a E. coli strain-specific effect, the authors should assess whether TIM-1 - mediated uptake is conserved across additional laboratory, K12, wild-type, pathogenic, or clinical E. coli strains.

Reviewer #2: (No Response)

Reviewer #3: Overall, the manuscript is well written and the data are support the conclusions. I have several comments that, if addressed, would further strengthen the study:

1. Additional detail regarding the screened receptor library would be helpful. How was the threshold for defining “significant binders” determined?

2. Only TIM-1 (and possibly TIM-4) were validated functionally in cells, whereas other hits from the screen did not alter OMV uptake upon overexpression. Are these additional candidates expressed at the cell surface in the tested epithelial cell lines? If not, this should be stated clearly.

3. Relatedly, surface expression of TIM-1 was assessed by flow cytometry, but the corresponding data are not clearly shown – the FACS data in the SI is not easy to interpret in terms of % surface expressed. Similar surface expression validation should be provided for other tested receptors.

4. OMVs were normalized by protein concentration (e.g., 50 µg/mL), but protein content can vary substantially depending on strain background and growth conditions. It would be helpful to provide particle number estimates (e.g., via nanoparticle tracking analysis) and, if possible, an approximate multiplicity of vesicles per cell. Relating the in vitro exposure levels to physiologically relevant concentrations encountered during infection would be helpful.

5. Please provide more detail on the LPS used in competition assays (Fig. 3). Was it derived from a strain expressing O-antigen or from a rough mutant lacking O-antigen? Because O-antigen is part of LPS, the wording in the manuscript describing “LPS and O-antigen” as separate surface components should also be revised for accuracy.

6. The TIM-1 overexpression and knockout data are convincing, but additional context would strengthen the physiological relevance. Can the authors comment on endogenous TIM-1 expression levels in the tested epithelial cell lines and how they compare to primary epithelial or immune cells in vivo? Since TIM-1 expression is often inducible under stress or injury conditions, this discussion would help frame the in vivo implications.

7. The experiment expressing TIM-1 in monocytes to assess cell death requires clearer justification. What is the relevance? Monocytes are known to internalize OMVs and mount inflammatory responses in the absence of TIM-1.

**Part III – Minor Issues: Editorial and Data Presentation Modifications**

Reviewer #1: While the authors underline the need to identify additional receptors, involvement of receptor-independent uptake mechanisms, such as direct membrane fusion, have been described and merit discussion.

The manuscript would benefit from a more comprehensive discussion of other putative OMV receptors including pattern recognition receptors (PRRs), particularly Toll-like receptors that detect M/PAMPs such as LPS, flagellin, and peptidoglycan, as well as membrane glycolipids such as ganglioside GM1 (the cholera toxin receptor) or globotriaosylceramide Gb3 (the Shiga toxin receptor)...

The first paragraph of the discussion resumes arguments already presented and extensively developed in the introduction and should be shortened or removed.

Reviewer #2: (No Response)

Reviewer #3: 8. The wording describing the OMV surface composition should be revised: O-antigen is part of LPS and should not be described as a separate surface component.

9. The pink and purple colors used in some graphs are difficult to distinguish.

10. Fonts in the Supplementary Information are inconsistent and should be standardized.

PLOS authors have the option to publish the peer review history of their article (what does this mean?). If published, this will include your full peer review and any attached files.

**Do you want your identity to be public for this peer review?** For information about this choice, including consent withdrawal, please see our Privacy Policy.

Reviewer #1: No

Reviewer #2: No

Reviewer #3: No

**Figure resubmission:**
---

## [Editor Report · Decision Letter 1]

13 May 2026

Dear Dr MacNair,

We are pleased to inform you that your manuscript 'Outer Membrane Vesicles Hijack TIM-1 for Cellular Uptake' has been provisionally accepted for publication in PLOS Pathogens.

Best regards,

Eric Oswald, Ph.D., D.V.M.

Academic Editor

PLOS Pathogens

Thomas Guillard

Section Editor

PLOS Pathogens

Sumita Bhaduri-McIntosh

Editor-in-Chief

PLOS Pathogens

orcid.org/0000-0003-2946-9497

Michael Malim

Editor-in-Chief

PLOS Pathogens

orcid.org/0000-0002-7699-2064
---

## [Editor Report · Acceptance letter]

Dear Dr MacNair,

We are delighted to inform you that your manuscript, "Outer Membrane Vesicles Hijack TIM-1 for Cellular Uptake," has been formally accepted for publication in PLOS Pathogens.

Best regards,

Sumita Bhaduri-McIntosh

Editor-in-Chief

PLOS Pathogens

orcid.org/0000-0003-2946-9497

Michael Malim

Editor-in-Chief

PLOS Pathogens

orcid.org/0000-0002-7699-2064